# The Time Course of MHC-I Expression in C57BL/6J and A/J Mice Correlates with the Degree of Retrograde Gliosis in the Spinal Cord following Sciatic Nerve Crush

**DOI:** 10.3390/cells11233710

**Published:** 2022-11-22

**Authors:** Bruno Henrique de Melo Lima, André Luis Bombeiro, Luciana Politti Cartarozzi, Alexandre Leite Rodrigues de Oliveira

**Affiliations:** Laboratory of Nerve Regeneration, University of Campinas—UNICAMP, Cidade Universitária “Zeferino Vaz”, Rua Monteiro Lobato 255, Campinas 13083-862, SP, Brazil

**Keywords:** axotomy, sciatic nerve, nerve regeneration, plasticity

## Abstract

The pleiotropic role of the major histocompatibility complex class I (MHC-I) reflects the close association between the nervous and immune systems. In turn, MHC-I upregulation postinjury is associated with a better regenerative outcome in isogenic mice following peripheral nerve damage. In the present work, we compared the time course of neuronal, glial, and sensorimotor recovery (1, 3, 5, 7, and 28 days after lesion—dal) following unilateral sciatic nerve crush in A/J and C57BL/6J mice. The A/J strain showed higher expression of MHC-I (7 dal, ** *p* < 0.01), Iba-1 (microglial reaction, 7 dal, *** *p* < 0.001), and GFAP (astrogliosis, 5 dal, * *p* < 0.05) than the C57BL/6J counterpart. Synaptic coverage (synaptophysin) was equivalent in both strains over time. In addition, mRNA expression of microdissected spinal motoneurons revealed an increase in cytoskeleton-associated molecules (cofilin, shp2, and crmp2, * *p* < 0.05), but not trkB, in C57BL/6J mice. Gait recovery, studied by the sciatic functional index, was faster in the A/J strain, despite the equivalent results of C57BL/6J at 28 days after injury. A similar recovery was also seen for the nociceptive threshold (von Frey test). Interestingly, when evaluating proprioceptive recovery, C57BL/6J animals showed an enlarged base of support, indicating abnormal ambulation postinjury. Overall, the present results reinforce the role of MHC-I expression in the plasticity of the nervous system following axotomy, which in turn correlates with the variable recovery capacity among strains of mice.

## 1. Introduction

The implication of the major histocompatibility complex class I (MHC-I) in the rewiring of the spinal cord following peripheral nerve lesion reinforced the close interconnection between the nervous and immune systems. The differential expression of MHC-I in response to interferon beta administration further showed that the regenerative potential of lesioned neurons can be enhanced by early retraction of presynaptic inputs [1].

MHC-I upregulation following axotomy has been shown in different instances, and comparison among mouse strains revealed an innate variation in MHC-I expression, which was higher in C57BL/6J mice than in 129/SvJ siblings [2]. In line with this, we have previously shown that A/J mice present the greatest postinjury expression of MHC-I among the studied strains, which correlates with the increase in astroglial and microglial reactions in the spinal cord after peripheral axotomy [3]. Such findings demonstrate that the genetic background contributes to the response to injury and the success of the regenerative effort that follows [4,5]. Interestingly, despite a common genomic origin, mouse strains vary in some aspects of their regenerative capacity in response to a peripheral lesion, such as crushing or experimental nerve section [3,6,7]. Among them, C57BL/6J animals have shown increased vulnerability when compared to other isogenic strains [8]. In facial nerve axotomy experiments, the C57BL/6J strain presented 35% neuronal survival, a value much lower than the 90% found in the A/J strain. After sciatic nerve transection, C57BL/6J mice display a significant loss of dorsal root ganglia (DRG) neurons [9], influencing the sensory regenerative outcome, which is decreased in comparison to the A/J counterpart [4].

The differences among mouse strains are not restricted to the survival of neurons. According to Lainetti et al. [4], after sciatic nerve axotomy followed by tubulization, C57BL/6J mice displayed significantly fewer myelinated axons than BALB/cJ, DBA/1J, C3H/HeJ, or A/J mice. To understand the origin of the variability in the regeneration process, Lu et al. [7] investigated the genetic basis of some inbred lines, and the results indicated that the delay in axonal regeneration in C57BL/6J animals is not related to allele changes in only one gene, increasing the variability of the mechanisms that possibly influence nerve regeneration.

Additionally, regarding non-neuronal cells, especially astrocytes, which are distributed around motoneurons in the spinal cord cortex and participate in the modulation of the neuronal microenvironment [10], it was observed that after peripheral injury, isogenic strains may show differences in the activation pattern (time distribution) and intensity (expression level) [3]. Microglia, with their basal role of surveillance and maintenance of local homeostasis, can present different activation behaviors after injury, branching out and adhering to proinflammatory and cytotoxic profiles (which can be deleterious to regeneration) or exhibiting a proregenerative profile (oriented to cell protection) [11]. Nevertheless, it has been shown that astrocyte and microglial reactions are interconnected, and do not influence, at least directly, synaptic pruning following axotomy. Berg et al. [12] have provided evidence that synaptophysin immunoreactivity is not influenced by the degree of astroglial and microglial reaction. Nevertheless, more recently, it has become clear that labeling intensity alone is not sufficient to predict the degree of activation of such glial cells, and phenotyping is necessary to investigate A1–A2 and M1–M2 spectral polarization [13,14].

Both the glial and neuronal responses after an injury have been studied in our group in association with MHC-I expression [1,15,16]. In this sense, it has been shown that MHC-I molecules in the nervous system are associated with the maintenance of inhibitory synapses, as demonstrated by Oliveira et al. [17]. Furthermore, the difference in MHC-I expression may be related to the variability of synaptic plasticity, after injury, seen in different strains of isogenic mice [3].

In addition to the MHC-I molecule, proteins such as cofilin are important because they are related to cell cytoskeleton dynamics and synapse remodeling. [18,19,20]. As an example, cortical and hippocampal neurons display reduced neurite formation in cofilin knockout mice [21]. On the other hand, tropomyosin receptor kinase B (TrkB) is a signal transduction receptor for neurotrophins [22]. It is known that the presence of TrkB is a pro-axonal growth stimulus, whereas in its absence, neurite outgrowth is inhibited [23]. Furthermore, TrkB downregulation increases microgliosis and cellular apoptosis, in turn decreasing local trophic support [24]. Additionally, TrkB is related to the protein phosphatase SHP-2 in the inhibition of axonal growth [19]. Other molecules, such as collapsin response mediator protein 2 (CRMP-2), seem to influence axonal growth, differentiation, and regeneration [25].

Therefore, in the present study, the time course of MHC-I expression in C57BL/6J and A/J mice was evaluated and correlated with the glial reaction and sensorimotor recovery. Additionally, the gene expression of key cytoskeleton-related molecules, namely, cofilin, Shp-2, and Crmp2, as well as the trkb neurotrophin receptor, was evaluated in laser-dissected axotomized motoneurons by qRT–PCR. The results indicate that A/J mice undergo more prominent MHC-I upregulation 7 days postinjury, which is in line with augmented astroglial and microglial reactions. C57BL/6J mouse motoneurons, in turn, show enhanced gene expression of cofilin, shp2, and crmp2, indicating compensatory pathways to achieve axonal regrowth and repair. Overall, both strains reached similar sensorimotor recovery 28 dal, although A/J mice displayed a faster initial improvement than C57BL/6J mice. In addition, the present results reveal that C57BL/6J mice display an increased base of support after lesion, indicating a persistent impediment following axotomy. The data herein reinforce that genetic differences among strains of mice may trigger different regenerative strategies postinjury, which may indicate that multiple approaches are necessary to achieve robust recovery after axotomy.

## 2. Materials and Methods

### 2.1. Animals

C57BL/6J-Unib mice (male, 6–8 weeks old, 15–20 g) and A/J-Unib (male, 6–8 weeks old, 15–20 g) were obtained from the Multidisciplinary Center for Biological Investigation from the University of Campinas (CEMIB/UNICAMP) and housed in the animal facility of the Laboratory of Nerve Regeneration, Institute of Biology, UNICAMP. The animals were kept under a light–dark cycle (12 h/12 h) with controlled temperature and humidity and with water and palletized food ad libitum. The experiment was approved by the Institutional Committee for Ethics in Animal Use (Institute of Biology—CEAU/IB/UNICAMP, protocol 4558-1/2017) and performed in accordance with the guidelines of the National Council for Animal Experimentation Control (CONCEA). In total, 83 mice were used; the experimental groups are detailed in Figure 1.

### 2.2. Surgical Procedures

The mice were anesthetized with inhaled isoflurane (3% induction), and the lateral face of the left hind limb was depilated. Bepanthen (Bayer, Germany) was applied over the eyes to prevent dryness. Then, they were gently placed on a heated pad (36 °C) and kept anesthetized during surgery with isoflurane (1–2%, inhaled). Surgical access was performed through an incision parallel to the femur. The biceps femoris muscle and fascia were divulged, revealing the sciatic nerve. The nerve crush was performed close to the greater sciatic notch using a beer clamp tool (Honer-Medizin, Germany) [26] with a constant pressure of 14 MPa for 30 s. The muscles were repositioned, the skin sutured, and the mice were kept under controlled heating until they had completely recovered from the anesthesia. Animals received tramadol hydrochloride (5 mg/kg, subcutaneously) every 8 h for 24 h for analgesia.

### 2.3. Tissue Collection and Preparation

The mice were anesthetized with an overdose of xylazine and ketamine. Then, they were subjected to a thoracotomy followed by transcardial perfusion with 0.1 M sodium-phosphate-buffered saline (SPBS, with 0.9% NaCl; pH 7.38). The animals used for immunohistochemistry were subsequently perfused with a fixative solution (4% paraformaldehyde in 0.1 M PBS, pH 7.38), and their lumbar intumescence was dissected, removed, and immersed in the same fixation solution overnight at 4 °C. Subsequently, the specimens were washed three times with 0.1 M PBS and immersed in sucrose solutions with increasing concentrations for 24 h each (10%, 20%, and 30%). Samples for immunohistochemistry were immersed in Tissue-Tek, frozen in n-hexane at a controlled temperature (−32 °C to −35 °C), and stored at −20 °C. In contrast, samples for qRT–PCR were not fixed, and remained stored in a freezer at −80 °C directly after perfusion of 0.1 PBS.

### 2.4. Immunohistochemistry

Transverse sections of lumbar intumescence, 12 µm thick, were obtained using a cryostat (Microm, HM525), transferred to gelatin-coated glass slides, and stored at −20 °C until use.

For immunohistochemistry, the slides were kept at room temperature, and the sections were outlined with a hydrophobic pen (PAP pen, Sigma Z377821). Then, the slides were transferred to a humid and light-protected chamber. Sections were immersed in 0.01 M PBS (3 × 5 min each), dried, and incubated in 150 µL of blocking solution (3% bovine serum albumin in 0.1 M PBS) for 45 min. After that, the primary antibodies (Table 1) were diluted in an incubation solution (1.5% bovine serum albumin and 0.2% Tween in 0.1 M PBS) and left immersed overnight at 5 °C.

After primary antibody incubation, the sections were washed with 0.01 M PBS and incubated at room temperature with the appropriate secondary antibody (Table 1) for 45 min. Sections were again washed with 0.01 M PBS, dried, and coverslipped with glycerin/PBS (3:1). The slides were observed with an epifluorescence microscope (Leica DMB5500) and documented using a digital camera (Leica DFC 345 FX) with specific filters, depending on the secondary antibodies.

For quantification, three representative images for each specimen in each experimental group per strain were selected. The integrated density of pixels, representing protein immunostaining intensity, was measured in the lateral motor nucleus of the ventral horn ipsilateral to the lesion, as previously described by Oliveira et al. [17], using ImageJ software (version 1.33 u, National Institutes of Health, Bethesda, MD, USA). For the synaptophysin marker, the contralateral side was also quantified. Animals without injury (unlesioned) were used as controls. The integrated density of pixels was acquired per animal, and the mean ± standard error of the mean (SEM) of each experimental group was calculated.

### 2.5. Morphological Evaluation of Microglial Reactions

For microglial morphological evaluation, Iba-1 staining images were used. Five images from each animal of the respective experimental group and time point were acquired at the ventrolateral nucleus of Rexed lamina IX.

The morphology of the microglia was determined using ImageJ software (National Institutes of Health, Bethesda, Maryland, MD, USA, 1.48v) by an experimenter blinded to the experimental conditions. The cells were classified into five types: type I, cells with one or two cellular processes; type II, cells with three to five short branches; type III, cells with more than five long branches and a small cell body; type IV, cells with a large soma and several thick and retracted processes; and type V, cells with ameboid soma and numerous small processes. Type I and II microglia were considered surveillant, while types III, IV, and V were considered as activated microglia.

### 2.6. Real-Time PCT (RT–qPCR)

The lumbar intumescence was frozen in Tissue-Tek, segmented in 12 µm thick cross-sections, and arranged on RNAse-free slides. The lamina IX motoneurons were individually microdissected (Laser Microdissection—PALM—Laboratory of Multiuser Equipment, Laboratory of Molecular Genetics of the School of Medical Sciences—FCM/UNICAMP). To visualize the nerve cells, staining was performed with a specific kit. After dissection, samples were subjected to a sequence of extraction buffer, conditioning buffer, water, and 70% ethanol, and were passed through purification columns (Arcturus^®^ HistoGene^®^ LCM Frozen Section Staining Kit, Thermo Fisher, Carlsbad, CA, USA, KIT 0401). During RNA isolation, DNase was treated according to the manufacturer’s instructions. Then, complementary DNA (cDNA) was synthesized from 20 ng of RNA (High-Capacity cDNA Reverse Transcription Kit, Applied Biosystems, Foster, CA, USA, catalog number: 4374966), and the isolated RNA (10 µL) was added to a mixture of reagents containing a reaction buffer, dNTPs, primers, an RNase inhibitor, reverse transcriptase, and water. The samples were transferred to a thermocycler (97 °C, 120 min) for cDNA synthesis. As the RNA value obtained by microdissection was low, pretreatment of the cDNA referring to the genes of interest was performed (Table 2) using a preamplification kit (TaqMan™ PreAmp Master Mix Kit, Vilnius, Lithuania, catalog number: 4384267). In summary, for each cDNA sample (12.5 μL), a mixture of equal volume was added, containing all primers (TaqMan assays) diluted in water, at a final concentration of 0.05× each. Then, 25 μL of TaqMan PreAmp Master Mix (2×) was added, and the samples were amplified (first cycle of 10 min, 95 °C, followed by 10 cycles of 15 s, 95 °C, 4 min, 60 °C).

Finally, the samples (in triplicate) were amplified through real-time RT–qPCR. For each sample, the preamplification product was added (1/2,5 diluted in water; 5 μL) to TaqMan assays (primer plus probe, 20×; 1 μL; Table 2), water (4 μL), and *TaqMan Gene Expression Master Mix 2x reagent* (*TaqMan™ Gene Expression Master Mix*, *Vilnius, Lithuania*, *catalog number*: *4369016*; 10 μL). The reaction took place in the thermocycler as follows: 1 °C, 10 min, 95 °C; next cycles, 10 s, 95 °C, 1 min, 60 °C. After analysis by the plug-in “*Best Keeper*” (*Biotechnology Letters*, *26:509-512*), GAPDH was chosen as the appropriate endogenous control. Quantitative PCR analysis was performed on the Mx3005 P system (Agilent, Santa Clara, CA, USA), and the results were calculated using the associated software (Agilent). The relative quantification of the genes of interest was calculated using the 2^−ΔΔCt^ method [27].

### 2.7. Functional Recovery (Motor Function, Proprioception, and Nociceptive Test)

The functional analysis of gait was performed using the automated walkway system for gait (Cat Walk System, Noldus Inc., Wageningen, The Netherlands), in which the animal walked on a glass platform illuminated by a green light, enhancing the footprint each time the paws came into contact with the floor of the equipment. Three runs were recorded by a high-speed camera positioned under the catwalk, and the data were stored using CatWalk software XT 10.5 (Noldus Inc.). The sciatic functional index (SFI) was calculated according to the formula of Inserra et al. [28]: SFI = 118.9 ((ETS–NTS)/NTS) − 51.2 ((EPL–NPL)/NPL)−7.5, where N is the normal side; E is the experimental side (crushed); PL is the print length; and e TS is the distance between the first and fifth fingers (toe spread). In addition, data referring to the base of support (BOS) of the pelvic limbs of the animals were evaluated as an indirect measure of proprioception. For adaptation, the mice were individually placed in the system to walk for 5 min, once a day, on three different days, before the beginning of the experiments. Two days before crushing of the sciatic nerve, the animals were placed on the platform, the values were acquired, and the best value obtained was used as a control. The SFI values were normalized as a function of those obtained in the preoperative period, and are expressed as a percentage.

Data were acquired again on the 4th day after the injury, and this was repeated every 4 days until the 4th week (28 days). For data collection, each animal was placed on the platform, free to walk in both directions, with valid runs, and captured only with a duration between 0.50 and 5.00 s and greatest speed variation of 40%. The camera gain was set to 25.01, and the detection limit was set to 0.25. Three runs following these criteria were acquired per trial, and no food restrictions or rewards were used.

The test for the nociceptive threshold was performed by analyzing the spinal reflex (mechanoreceptive reflex arc) through an electronic esthesiometer meter (von Frey test) for 28 days at intervals of 3 to 4 days. The plantar region of the left hind paw was stimulated with the tip of a pipette coupled to a force transducer connected to a digital potentiometer that calculated the applied pressure, in grams, necessary to invoke the reflex arc of paw withdrawal. The precision of the device was 0.5 g, which was calibrated to receive a maximum force of 15 g; however, so that there was no risk of tissue damage to the animals, the maximum limit of applied pressure was 8 g. If the animal did not respond to this intensity of pressure, paw anesthesia was considered. The paw withdrawal reflex threshold was obtained before and after sciatic nerve crush for both strains, with the results presented as the mean ± standard error of the mean (SEM) of the data obtained for each group of animals. The stimulus was repeated a maximum of six times until four consecutive consistent responses were generated per animal, according to the methodology established by Benitez et al. [29]. The results were normalized to the average force registered for each animal before the injury.

### 2.8. Statistical Analysis

All data are expressed as the mean ± SEM, and *p* < 0.05 was considered significant. The statistical analysis of the data was performed using GraphPad Prism (GraphPad Software, version 8.0.1, La Jolla, CA, USA). For the quantification of immunostaining and the behavioral test between animals of the same lineage, but in different groups (periods), a one-way analysis of variance (ANOVA) with repeated measures was performed, followed by the Bonferroni post hoc test for multiple comparisons. Two-way ANOVA followed by the Sidak and Bonferroni post hoc test was used to compare the different strains. PCR quantification and microglial morphological analysis were performed using an unpaired *t*-test.

## 3. Results

### 3.1. MHC-I Labeling Was Higher in the A/J Strain Than in the C57/BL6J Strain on the 7th Day after Injury (Figure 2)

Although basal MHC-I immunolabeling was minimal in both mouse strains, after crushing of the sciatic nerve, C57BL/6J animals displayed crescent upregulation until the 5th day after injury, when there was a robust increase in MHC-1 expression compared to the control group (unlesioned vs. 5 dal, ** *p* < 0.01), reaching its peak within the first week at 7 dal (unlesioned vs. 7 dal, **** *p* < 0.0001).

**Figure 2 cells-11-03710-f002:**
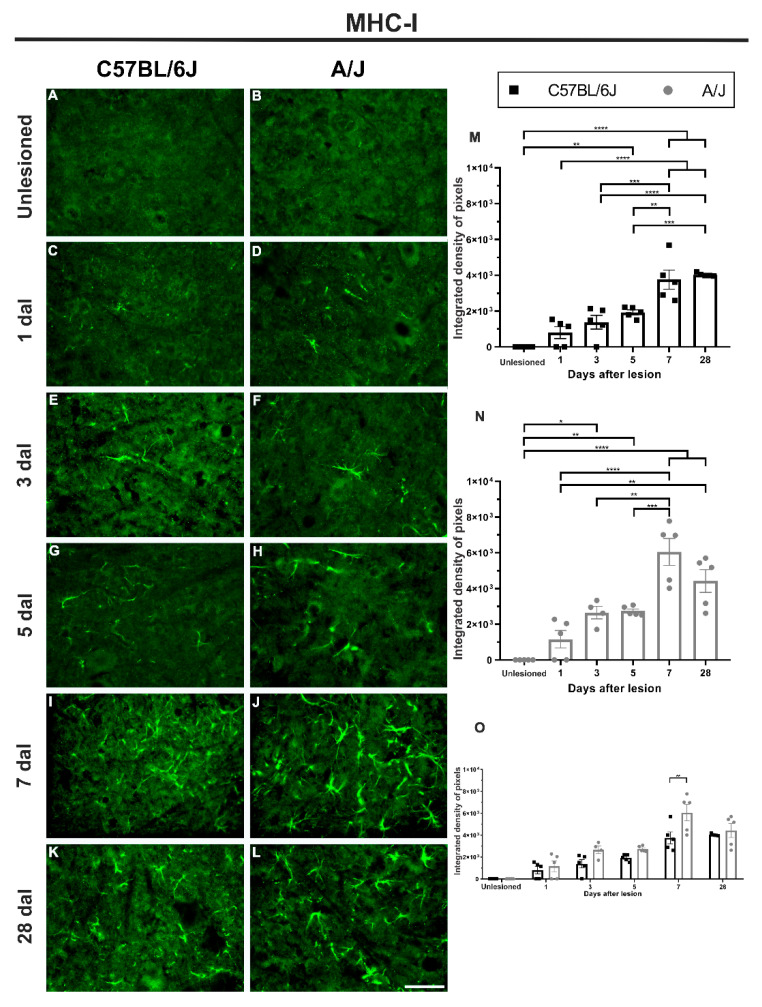
Immunostaining for MHC-I increases for both C57BL/6J and A/J strains after sciatic nerve crushing, with earlier and higher expression in the A/J strain when compared to C57BL/6J. (**A**–**L**) Representative images of lamina IX of Rexed from control animals (unlesioned) and in groups after peripheral nerve crush on Days 1, 3, 5, 7, and 28, ipsilateral to the lesion (with the exception of the control), in both strains. (**M**) MHC-I expression in C57BL/6J animals, showing its highest value within the first week, 7 days after lesion (dal), which remained upregulated at 28 dal. (**N**) MHC-I expression in A/J animals, with upregulated labeling at 3 dal, peaking at 7 dal, and remaining increased at 28 dal. (**O**) Comparison between the values found in the C57BL/6J and A/J strains on the different days after peripheral nerve injury, with a difference between the strains at 7 dal, being higher in the A/J mice. Quantification of the integrated density of pixels within lamina IX of Rexed (*n* = 5 per group/day; mean ± SEM; * *p* < 0.05; ** *p* < 0.01; *** *p* < 0.001; **** *p* < 0.0001). Scale bar: 50 μm.

In A/J mice, MHC-I upregulation began earlier than in C57BL/6J mice at 3 dal, with a significant increase relative to the control group (unlesioned vs. 3 dal, * *p* < 0.05), increasing at 5 dal (unlesioned vs. 5 dal, ** *p* < 0.01), and reaching peak expression within the end of the first week postinjury at 7 dal (unlesioned vs. dal, **** *p* < 0.0001). When comparing both strains at 7 dal, a significant difference was observed, such that the MHC-I expression in A/J mice was 1.6 times that in their C57BL/6J counterparts (** *p* < 0.01). At 28 dal, both strains presented elevated values compared to the control group.

### 3.2. The Synaptic Retraction Time Course following Sciatic Nerve Crush Is Equivalent in C57BL/6J and A/J Mice

Synaptophysin downregulation following axotomy allows the evaluation of synaptic retraction within the spinal cord, which is a hallmark of the retrograde response. C57BL/6J and A/J animals displayed a progressive decrease in synaptic labeling up to the 7th day after lesion (7 dal). C57BL/6J mice showed a decrease in synaptophysin expression of approximately 24% on the first day after injury (unlesioned vs. 1 dal, *** *p* < 0.001), with further loss of 27% and 32% (unlesioned vs. 3 and 5 dal, **** *p* < 0.001) on subsequent days, reaching a loss of 49% on 7 dal (unlesioned vs. 7 dal, **** *p* < 0.0001; Figure 3). Despite the partial synaptophysin immunostaining recovery over time, which reached approximately 80% of the initial value, the 28 dal labeling was inferior to the unlesioned counterpart (unlesioned vs. 28 dal, ** *p* < 0.01).

In A/J mice, despite showing a trend toward a gradual decrease in synaptic reactivity from the first day after lesion, it reached statistical significance at 3 dal, with a decrease of 23% (unlesioned vs. 3 dal, * *p* < 0.05), and remained so until 5 dal (unlesioned vs. 5 dal, * *p* < 0.05). The minimum integrated density of pixels was observed at 7 dal, with a decrease of 40% (unlesioned vs. 7 dal, *** *p* < 0.001). However, at 28 dal, no significant difference in comparison to the unlesioned counterpart was observed, in contrast to the C57BL/6J results.

### 3.3. The Microglial Reaction of the C57BL/6J Strain Precedes That of the A/J Mice but Is Lower in Intensity (Figure 4)

The C57BL/6J strain showed an immediate response to peripheral injury in the spinal cord microenvironment, with a 192% increase at 1 dal (unlesioned vs. 1 dal, * *p* < 0.05) and progressively increasing to 249% and 344% at 3 and 5 dal, respectively (unlesioned vs. 3 dal and 5 dal, ** *p* < 0.01 and *** *p* < 0.001), reaching peak intensity at 7 dal, with a reactivity of approximately 512% greater than that of the unlesioned group (unlesioned vs. 7 dal, **** *p* < 0.0001).

**Figure 4 cells-11-03710-f004:**
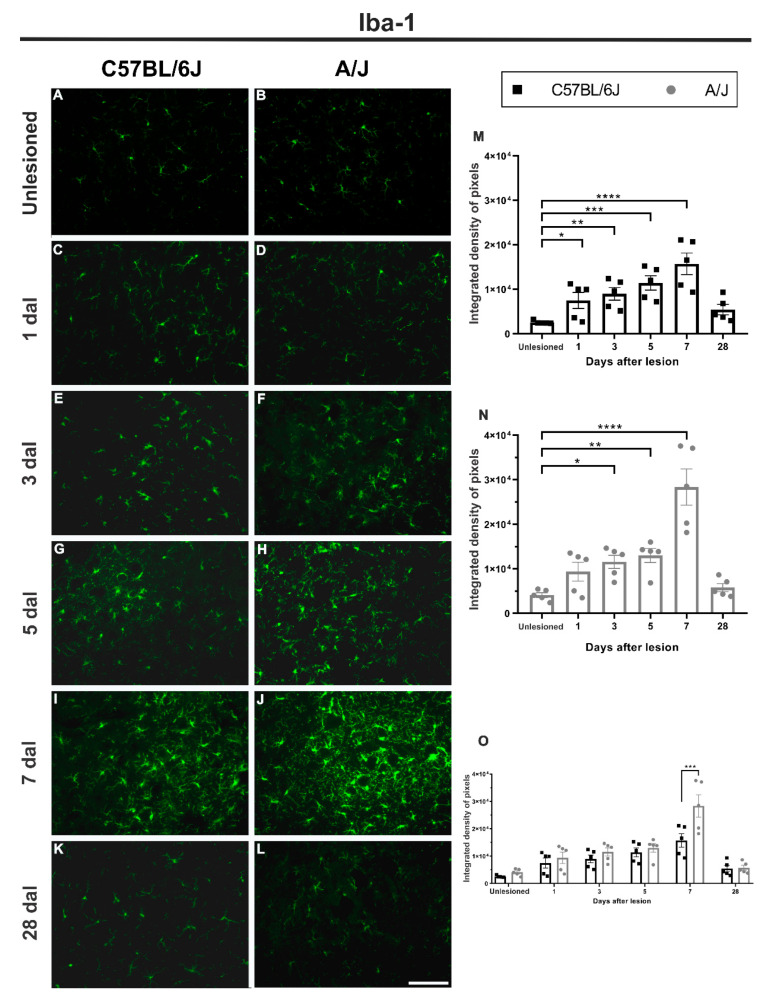
The microglial reaction (Iba-1 immunostaining) increased over time, with earlier upregulation in the C57BL/6J strain after sciatic nerve crush and with a peak at 7 days in both strains, being higher in the A/J strain. (**A**–**L**) Representative images of lamina IX of Rexed from control animals and from groups that received peripheral nerve crush on Days 1, 3, 5, 7, and 28, ipsilateral to the lesion (except the control), in both strains. (**M**) Microgliosis in C57BL/6J animals, with a progressive increase in Iba-1 staining from 1 to 7 dal. At 28 dal, the microglial reaction decreased, equaling the value found in the control group. (**N**) Reactive microgliosis in A/J animals, with a progressive increase in Iba-1 staining from 3 to 7 dal, within the studied intervals, the latter being the highest value found; at 28 dal, microgliosis decreased, equaling the value found in the control group. (**O**) Comparison between the values found in the C57BL/6J and A/J strains throughout the time course after peripheral nerve injury, with a significant difference between the strains on Day 7. Quantification of the integrated density of pixels within lamina IX of Rexed (*n* = 5 per group/day for each strain). Mean ± SEM; * *p* < 0.05; ** *p* < 0.01; *** *p* < 0.001; **** *p* < 0.0001. Scale bar: 100 μm.

A/J mice displayed a similar Iba-1 expression time course compared to C57BL/6J; although the upregulation became statistically significant from 3 dal, with an increase of approximately 180% relative to the unlesioned group (unlesioned vs. 3 dal, * *p* < 0.05). At 5 dal, the reactivity increased to 216% (unlesioned vs. 5 dal, ** *p* < 0.01), reaching its maximum value, similar to the C57BL/6J strain, on the 7th day, with microgliosis 589% higher than that of the control group (unlesioned vs. 7 dal, **** *p* < 0.0001). Of note, up to 7 dal, A/J mice displayed twice the microglial reactivity identified in the C57BL/6J lineage (A/J—7 dal vs. C57BL/6J—7 dal, *** *p* < 0.001). At 28 dal, both strains returned to basal levels of Iba-1 immunostaining.

### 3.4. Morphological Characterization of Microglial Reactions (Figure 5)

Complementary to the Iba-1 integrated density of pixels measurement, a quantitative assessment of microglial morphology was conducted. In the different strains and at different time points, we quantified the number of microglial cells, considering types I and II as surveying, or not activated, and types III, IV, and V as activated.

**Figure 5 cells-11-03710-f005:**
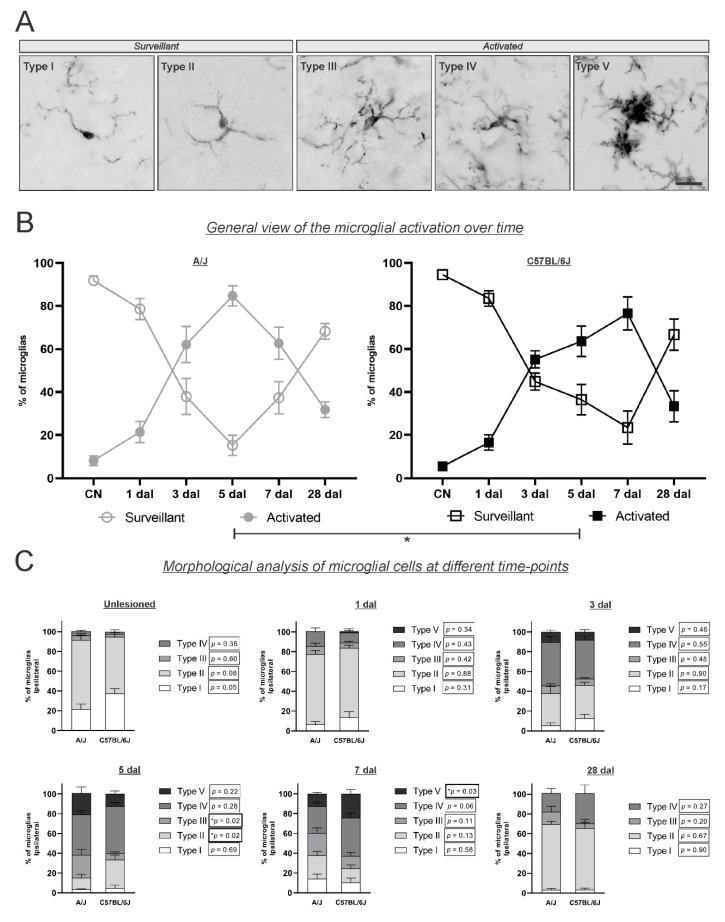
Morphological analysis of microglial activation in the A/J and C57BL/6J strains over time. (**A**) Representative photomicrographs of Iba-1 immunostained microglia classified, according to their morphology, into surveillant (types I and II) and activated microglia (types III, IV, and V). Scale bar: 10 μm. (**B**) Quantification of microglial cells in the surveillant and activated categories in the different experimental groups over time. Note that there was an increase in the activation percentage after the lesion, occurring in a time-dependent manner, which peaked in the A/J strain at 5 dal, earlier than in the C57BL/6J strain (* *p* = 0.03). (**C**) Detailed morphological typing of microglial cells or each experimental group and time point. Data were analyzed by comparing the two strains with an unpaired *t*-test at each time point. Note the higher percentage of type III microglia (the less activated morphology) in the A/J strain at 5 dal than in the C57BL/6J strain (* *p* = 0.02), and the increased level of type V microglia (the most activated morphology) in the C57BL/6J strain at 7 dal compared to the A/J strain (* *p* = 0.03).

In the unlesioned mice of both strains, approximately 95% of microglia displayed the surveying types of morphology (types I and II), with no significant difference. At 1 dal, 21% and 16% of microglia were already activated in A/J and C57Bl/6J mice, respectively, showing an increased number of type IV microglia (corresponding to approximately 12% of counted cells). At 3 dal, the microglial reaction was more intense; 62% and 55% of microglia were activated in A/J and C57Bl/6J mice, respectively, with a high detection of type IV (approximately 40%), and approximately 9% of the counted cells were characterized as type V microglia (the more activated phenotype) in both strains.

On the 5th day after lesion, a relevant difference in the activation pattern was detected among the strains. While 85% of the microglia were activated in the A/J strain, the highest activation level was found in this strain, and 64% of the microglial cells were activated in the C57BL/6J strain (*p* = 0.03). Of the different activated morphologies, there were more type III microglia in the A/J strain (*p* = 0.02). Unlike the A/J strain, the microglial activation peak in the C57BL/6J strain was found on the 7th day postlesion, when 77% of microglial cells were activated. In this stage, 25% of the microglial cells were characterized as type V, the most activated morphological type, in the C57BL/6J strain (*p* = 0.03).

At the latest analyzed time point, 28 dal, in both strains, the percentage of surveillant microglia was prevalent (68%), although it did not reach the unlesioned levels.

### 3.5. Reactive Astrogliosis in C57BL/6J Mice Precedes the A/J Response and Persists Longer, Becoming Equivalent at Peak Upregulation

Both strains showed progressive GFAP expression after sciatic nerve crush. This increase, in C57BL/6J animals, was statistically significant at 3 dal, being 113% higher than that found in the control group (unlesioned vs. 3 dal, * *p* < 0.05). The same was observed at 5 dal (unlesioned vs. 5 dal, * *p* < 0.05) (Figure 6).

In A/J mice, GFAP upregulation became significant at 5 dal, being 121% higher than the control group (unlesioned vs. 5 dal, * *p* < 0.05). At 7 dal, the astroglial reaction began to return to basal levels, different from the C57BL/6J animals that still showed higher reactivity, 65% greater than the control (unlesioned vs. 7 dal, * *p* < 0.05). At 28 dal, both strains returned to basal levels of GFAP expression.

### 3.6. Spinal Cord Astrocytes Express MHC-I, Unlike Microglia

Double labeling was performed for the anti-MHC-I antibody and astrocytic (GFAP) or microglial (Iba-1) cells (Figure 7A–H) to identify which cell type is responsible for the expression of this molecule related to the major histocompatibility complex on the 7th day after injury in both strains.

In C57BL/6J animals, when the markings are superimposed, it can be observed that astrocytic cells colocalize with the expression of MHC-I, both in their body and in their extensions (Figure 7A,B), unlike microglial cells (Figure 7C,D). The same phenomenon can be identified in the A/J lineage (Figure 7E–H).

### 3.7. C57BL/6J Mice Express Genes Involved with Axonal Cytoskeleton More Intensely and Persistently Than A/J Animals

Motoneurons from the C57BL/6J and A/J strains were microdissected at 7 dal based on the data indicating the greatest synaptic retraction combined with glial reaction (Figure 8). The expression of genes related to the axonal cytoskeleton was analyzed, and interestingly, only C57BL/6J mice showed a significant increase when compared to control animals for genes such as crmp2 (unlesioned vs. 7 dal, * *p* < 0.05), cofilin (unlesioned vs. 7 dal, * *p* < 0.05), and shp2 (unlesioned vs. dal, ** *p* < 0.01). In contrast, trkb expression was not altered in either strain 7 days after lesion induction. Of note, the expression of all the genes studied was higher in animals of the C57BL/6J strain when compared to A/J (C57BL/6J vs. A/J—crmp2, ** *p* < 0.01; cofilin, ** *p* < 0.01; shp2, * *p* < 0.05, and trkb, * *p* < 0.05). All data are detailed in Table 3.

### 3.8. Functional Recovery Measured by Gait, Proprioception and Sensory Response Were Fully Achieved in C57BL/6J and A/J Strains after Sciatic Nerve Crushing

The analysis of gait recovery in the CatWalk system, through the sciatic functional index, in the C57BL/6J and A/J mice was calculated every 4 days from before the sciatic nerve was crushed until 28 days after the operation (Figure 9A). There was no statistically significant difference between the strains, although A/J mice fully recovered by the 12th day after lesion, while the C57BL/6J strain showed prelesion results by the 20th day postinjury.

In C57BL/6J animals, a significant change in gait was observed until the 8th day (**** *p* < 0.0001) following injury, with a progressive improvement on the 12th, 16th, and 20th days (*** *p* < 0.001; ** *p* < 0.01; * *p* < 0.05, respectively). At 24 days after the injury, there was no significant difference relative to the control. The A/J animals also showed significant loss of SFI up to the 8th day postinjury (**** *p* < 0.0001). However, A/J mice showed a rapid improvement by the 12th day, which was already statistically similar to the preinjury values. Interestingly, on the 16th day, there was a decline in the SFI (* *p* < 0.05), which then recovered on the 20th day after injury. Therefore, at the endpoint of the study, the animals from both strains showed equivalent SFIs, compatible with normality.

The analysis of the nociceptive threshold using the von Frey test (Figure 9B) was applied to the C57BL/6J and A/J animals over 28 days at intervals of 3 to 4 days. Both strains showed similar patterns of analgesia/hypoalgesia or the motor impossibility of removing the paw after a noxious stimulus on the 4th and 7th days after peripheral nerve crush. On the 11th day, the animals began to react to the stimulus, following a similar pattern of hyperalgesia that continued until the 18th day. Interestingly, on the 21st day postinjury, there was a statistically significant difference between strains, with the A/J animals showing a peak sensitivity that was not seen in the C57BL/6J counterpart (A/J vs. C57BL/6J, * *p* < 0.05). On the 28th day of recovery, both strains returned a response similar to that detected before nerve injury.

Proprioceptive recovery was performed through the base of support (BOS) evaluation, comparing C57BL/6J and A/J mice (Figure 9C). Data were obtained every 4 days from preoperative analysis and after sciatic nerve crush until 28 dal.

In the C57BL/6J strain (Figure 9D), neglect of the injured limb was observed until the 8th day (**** *p* < 0.0001). After this time point, support was recovered, but with an increase in BOS, when compared to the control, on days 24 and 28 (* *p* < 0.05). The A/J mice (Figure 9D) also showed monoplegia of the injured limb until the 8th day. By 12 dal, however, the BOS did not present a significant difference relative to the control. Interestingly, at 20 dal, the A/J animals showed an increased BOS (* *p* < 0.05), normalized afterward. Importantly, when comparing the BOS time course, C57BL/6J mice showed higher numbers than the A/J siblings (Figure 9D).

## 4. Discussion

Peripheral nerve injuries are common events, and result in major sensorimotor dysfunctions. Due to the complexity of the nervous system’s response to injury, several parameters need to be taken into account when evaluating functional recovery. The level of the lesion is of relevance because the more proximal the axotomy is, the more intense the retrograde reaction, including neuronal degeneration and synaptic loss, as well as the glial reactions within the spinal cord. [30,31,32]. The age at injury also plays a major role in the prognosis of recovery. Thus, neonatal nerve trauma results in extensive retrograde motoneuron and sensory neuron degeneration, reducing sensorimotor recovery [33,34,35,36]. The genetic background is another parameter that still lacks investigation, although it is clear that there may be substantial differences in the response to lesions among individuals. This is apparent when comparing different mouse strains following axotomy [3,4,5,6,7], and has been investigated over the years.

C57BL/6J animals are known for having a low regenerative potential when compared to derived strains (such as the A/J strain) [4]. This has been shown to be a condition of polygenic influence and, therefore, to involve a complex number of pathways [7]. In this regard, Sabha et al. [3] focused on MHC-I expression and synaptic plasticity 1 and 3 weeks after sciatic nerve transection in different mouse strains, namely, C57BL/6J, A/J, and BALB/cJ. The present results differ from the previously published comparison among strains due to the time course of evaluation as well as the nature of the lesion, which in the current work is a crushing, instead of a cutting, of the sciatic nerve.

Thus, the findings herein reinforce that MHC-I upregulation in both C57BL/6J and A/J mice peaks at the 7th day postinjury, and is significantly higher in the A/J strain. Nonetheless, such differential MHC-I expression did not result in enhanced synaptophysin downregulation, which showed equivalent ratios in both strains. Thus, following crush injury, the similar presynaptic plasticity in both strains indicates a comparable regenerative outcome, which was observed up to 28 days of recovery by the behavioral evaluations. In fact, the recovery of C57BL/6J mice in the context of a Sunderland III lesion is similar to that of their A/J counterparts, both in terms of sensory feedback (von Frey flinch withdrawal test) and motor recovery (CatWalk—walking track test). Such contrast to the section of the nerve (Sunderland V) highlights the importance of the retrograde response to injury, proposed by Fawcett [37]. Of note, however, is the observation that the base of support recovery in C57BL/6J mice is significantly wider than that in A/J mice, indicating that proprioceptive inputs (Ia and II fibers) do not recover properly after nerve transection, which is possibly related to the MHC-I-driven plasticity of presynaptic inputs to alpha motoneurons [38,39,40].

The abovementioned alterations in the soma of motoneurons in response to the injured nerve, in addition to extensive synaptic retraction, elicit microgliosis and reactive astrogliosis [41]. This glial reaction has been extensively related to the plasticity of the nervous system, although this may occur indirectly by means of secretion of pro- and anti-inflammatory molecules. In fact, Berg and colleagues have observed that glial reactions around motoneurons do not influence the degree of synaptic plasticity, indicating that the retraction of presynaptic terminals is a postsynaptic neuronal autonomous response to axotomy. If so, the degree of injury should influence the outcome of the response.

Astrocytes, in particular, exert an indispensable influence on the stabilization and maintenance of synaptic contacts, enabling an adequate microenvironment around functional synapses [42,43,44] and limiting/modulating ions and neurotransmitters [45,46].

MHC-I expression is observed simultaneously in astrocytes and microglia, which is in line with experimental exogenous stimulation by interferon beta [1]. Herein, we show a more frequent colocalization of MHC-I immunolabeling with GFAP, indicating that astrocytes may respond more promptly to axotomized motoneuron signaling postinjury. Furthermore, an association between the degree of glial reactivity and synaptic elimination has been suggested [41,47,48,49,50]. According to the above discussion, the contrast between the cutting and crushing of the sciatic nerve may contribute to the astroglial differences detected when comparing previous data. Importantly, in the present work, clear behavioral differences could be observed in terms of motor coordination, reinforcing the correlation between the glial reaction and synaptic plasticity [3].

The upregulation of cofilin, Trkb, Shp2, and Crmp2 gene expression, particularly in C57BL/6J mice, is a key finding of the present study. The laser dissection of spinal motoneurons in the context of sciatic nerve injury is a novel approach that revealed some of the retrograde reactions of the axotomized neurons. To date, most of the data are the results of ventral horn, or even entire spinal cord, gene expression evaluation. Thus, cofilin upregulation in C57BL/6J motoneurons may be interpreted as a proregenerative effort, because this molecule relates to the formation of the axonal cytoskeleton. Interestingly, we proposed in a previous set of experiments that C57BL/6J lower regenerative potential may be due to a mismatch in the timing of neurotrophic stimuli and the ability of the neurons to regrow their axons [51]. In line with this, Frendo et al. [52] demonstrated that the presence of activated cofilin at the site of injury correlates with increased sprouting. The presence of Trkb sustained mRNA levels in C57BL/6J mice can also be associated with a compensatory response to axotomy. Regarding SHP-2, the study carried out by Kusakari et al. [53] indicates an association with TrkB, in which SHP-2 knockout mice show decreased TrkB effectiveness. Therefore, the upregulation of Shp-2 may in turn enhance TrkB activation, improving the sensitivity to BDNF.

Finally, the presence of CRMP-2, both in the developmental and adult CNS, appears to regulate neurite outgrowth. For example, Arimura et al. [54] showed the participation of CRMP-2 in vesicular anterograde transport through the axons of TrkB-positive neurons.

Overall, based on the results herein, C57BL/6J and A/J mice may respond differently to axotomy due to intrinsic metabolic variations that result in differential regenerative strategies. As a result, the regenerative outcome seems to be better in A/J mice, which respond faster and more effectively to the injury. C57BL/6J animals, as seen in several studies, show a substantial delay in the peripheral regenerative process, especially in more severe lesions, such as nerve transection. Even in the event of a milder lesion, C57BL/6J mice display greater fragility, which results in the incomplete recovery of sensorimotor function. This disadvantage can be triggered by the lower expression of MHC-I combined with slower activation of the glial response in the spinal cord microenvironment. Further studies will be necessary to evaluate whether such mechanisms can be modulated to improve regeneration postinjury.

## Figures and Tables

**Figure 1 cells-11-03710-f001:**
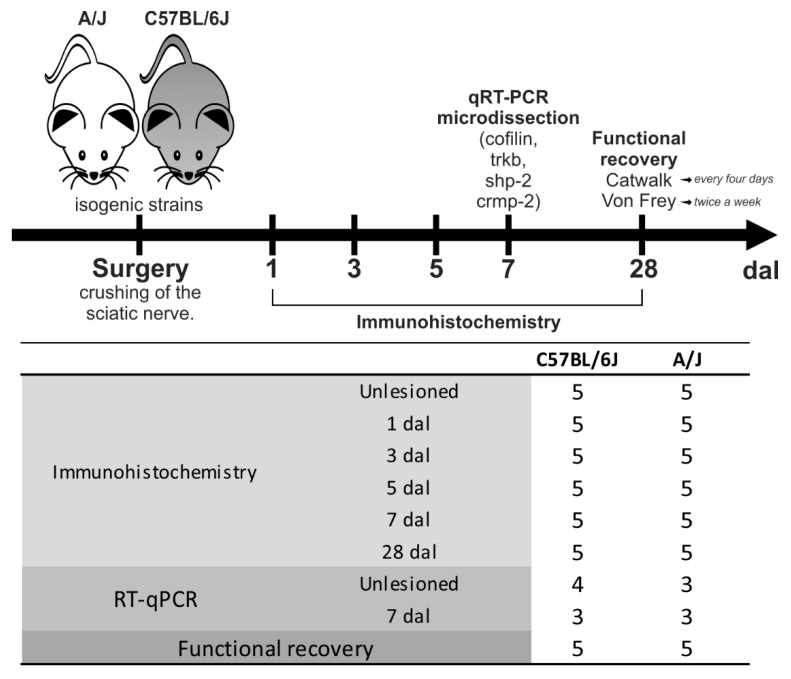
Schematic drawing and detailed distribution of animals used for experimentation, ordered by strain, according to the technique applied. For immunohistochemistry and RT–qPCR, groups with different survival periods after sciatic nerve crush were analyzed. Uninjured animals were used as controls. Dal: days after lesion.

**Figure 3 cells-11-03710-f003:**
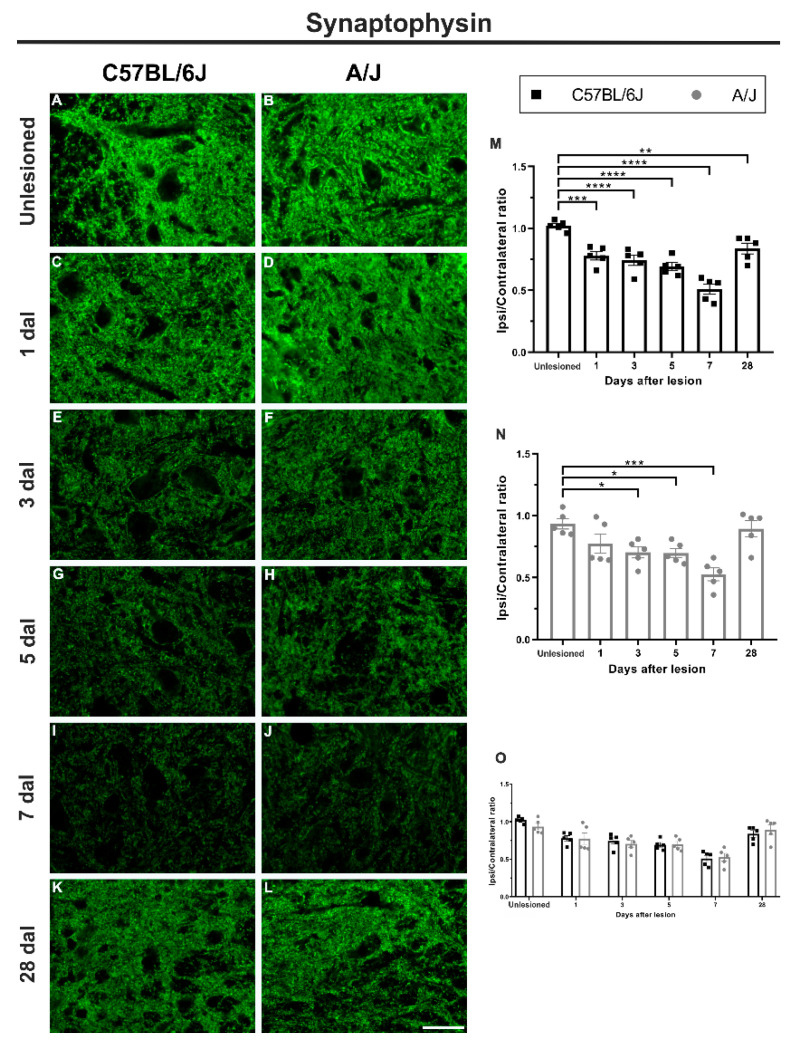
Synaptophysin immunostaining (presynaptic marker) decreased over time, with the lowest value 7 days after sciatic nerve crush in the C57BL/6J and A/J strains. (**A**–**L**) Representative images of lamina IX of Rexed in control animals and in the groups that received peripheral nerve crush 1, 3, 5, 7, and 28 days after lesion, both ipsilateral and contralateral to injury. (**M**) Synaptic reactivity in C57BL/6J animals, with a progressive decrease in synaptophysin labeling from 1 to 7 days after lesion (dal), at which the lowest value was reached. After 28 d, synaptophysin expression improved, but was still lower than that found in the control group. (**N**) Synaptic labeling in A/J mice, with a progressive decrease in the labeling for synaptophysin 3 to 7 dal, reaching the lowest value. Note that at 28 dal, synaptophysin expression improved, corresponding to the value found in the control group. (**O**) Comparison between the values found in the C57BL/6J and A/J strains on the different days after peripheral nerve injury, with no significant difference between the strains. Quantification of the integrated density of pixels was carried out on the surface of the motoneurons (ipsilateral/contralateral ratio, *n* = 5 per group/dal). Mean ± SEM; * *p* < 0.05; ** *p* < 0.01; *** *p* < 0.001; **** *p* < 0.0001. Scale bar: 50 μm.

**Figure 6 cells-11-03710-f006:**
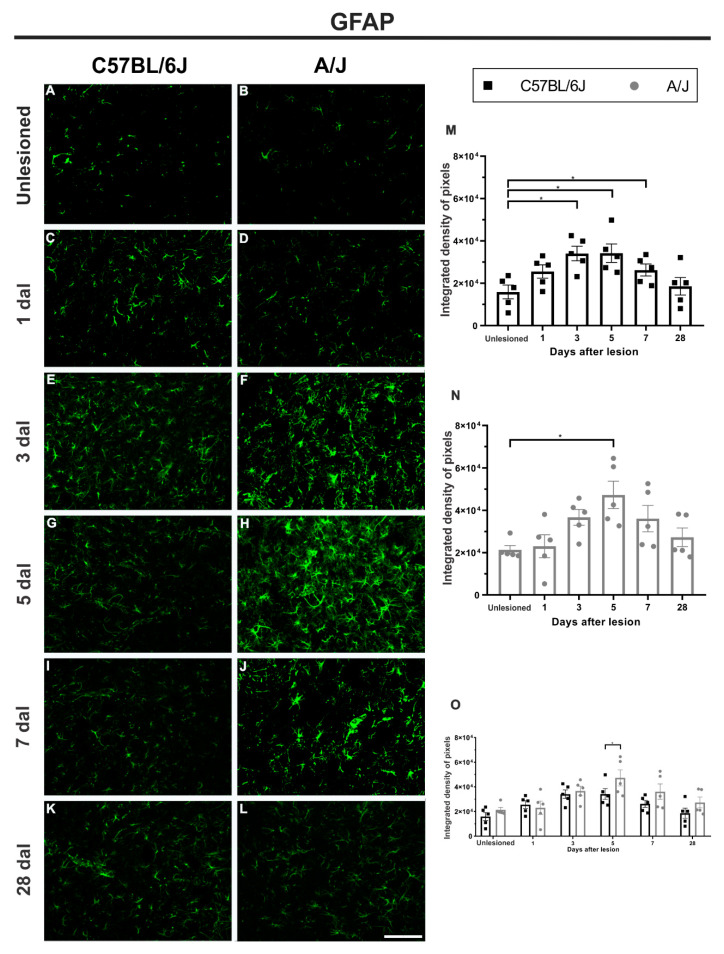
Reactive astrogliosis (GFAP immunostaining) increased over time in C57BL/6J and A/J mice after sciatic nerve crush, displaying an equivalent peak between strains at 5 days, but with earlier expression in C57BL/6J. (**A**–**L**) Representative images of lamina IX of Rexed from control animals and from groups that received peripheral nerve crush on Days 1, 3, 5, 7, and 28, ipsilateral to the lesion (except the control), in both strains. (**M**) Reactive astrogliosis in C57BL/6J animals, with a progressive increase in GFAP staining from 3 to 5 dal, the latter being the highest value found within the studied intervals; at 7 dal, a decrease in expression is observed, reaching the basal level at 28 dal. (**N**) Reactive astrogliosis in A/J animals, with a progressive increase in GFAP staining, although significant only at 5 dal; at 28 dal, astrogliosis decreases, equaling the value found in the control group. (**O**) Comparison between the values found in the C57BL/6J and A/J strains on the different days after peripheral nerve injury, indicating a difference between the strains on the 5th day (* *p* < 0.05). Quantification of the integrated density of pixels within lamina IX of Rexed (*n* = 5 per group/day for each strain). Mean ± SEM; * *p* < 0.05. Scale bar: 100 μm.

**Figure 7 cells-11-03710-f007:**
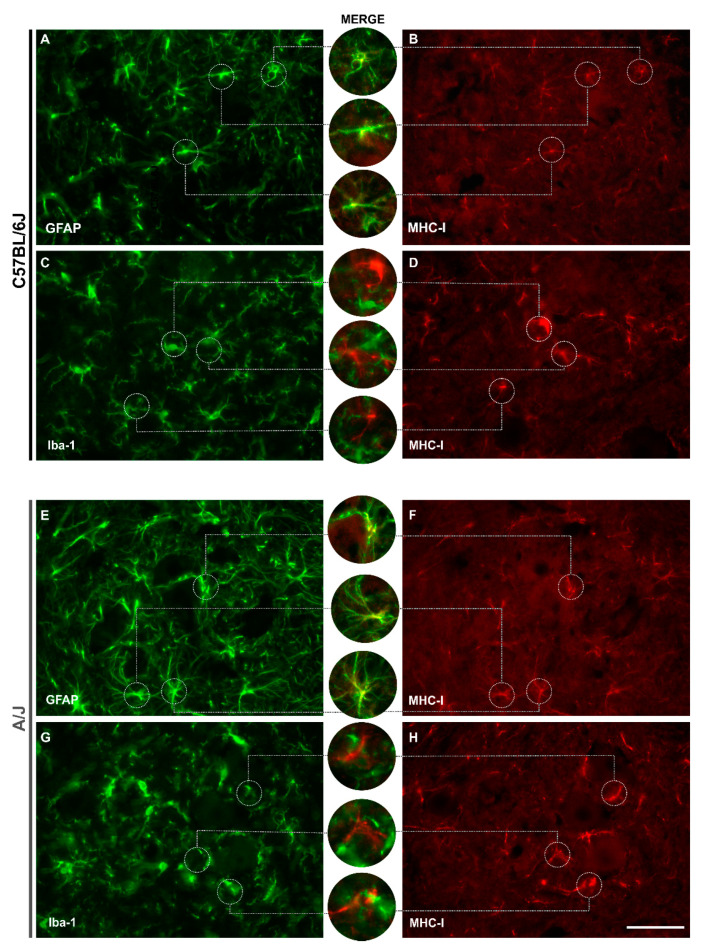
Double labeling for major histocompatibility complex class I (MHC-I) and astrocytic (GFAP) or microglial (Iba-I) cells in C57BL/6J and A/J strains 7 days after injury (7 dal). In the C57BL/6J mice, colocalization is observed between the anti-GFAP and MHC-I antibodies (**A**,**B**), which does not occur in the anti-MHC-I and anti-Iba-I double staining analysis. (**C**,**D**). The same pattern was observed in the A/J mice, both for astrocytes (**E**,**F**) and for microglia (**G**,**H**). Scale bar: 50 μm.

**Figure 8 cells-11-03710-f008:**
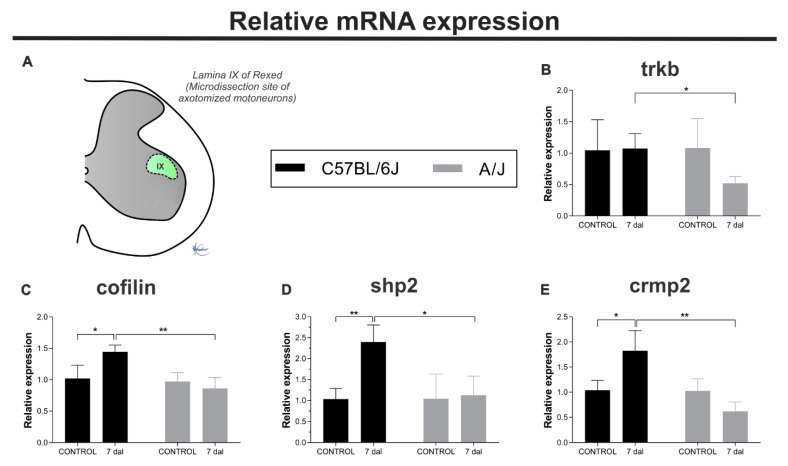
Relative expression of axonal cytoskeleton-related genes in microdissected motoneurons present in the dorsal lamina IX of Rexed (**A**) in control animals and 7 days after sciatic nerve crush. (**B**) trkb mRNA levels, (**C**) cofilin mRNA levels, (**D**) shp2 mRNA levels, and (**E**) crmp2 mRNA levels. Mean ± SEM (* *p* < 0.05; ** *p* < 0.01).

**Figure 9 cells-11-03710-f009:**
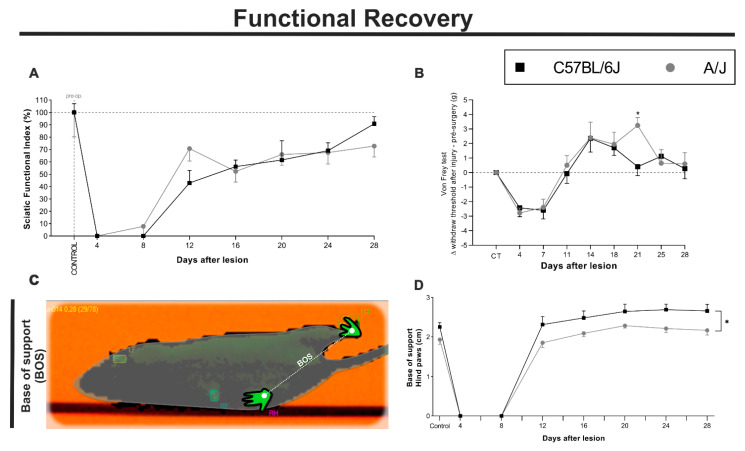
Functional recovery of C57BL/6J and A/J animals after crushing of the sciatic nerve through the sciatic functional index—SFI and the BOS (CatWalk system, motor analysis) in addition to the nociceptive threshold (von Frey system, sensory analysis). (**A**) SFI was analyzed in the C57BL/6J and A/J strains for 4 weeks (28 days). Mice from both strains fully recovered their gait, obtaining values similar to those found in the preoperative evaluation. No significant differences could be obtained. (**B**) The nociceptive threshold was analyzed in the C57BL/6J and A/J strains for 4 weeks (28 days) at intervals of 3 to 4 days. It was observed that the animals in both strains recovered sensitivity, obtaining values similar to those found in the preoperative evaluation, and that when the nociceptive threshold was compared between both strains, A/J mice presented a peak in hypersensitivity on the 21st day of recovery (21). (**C**) Schematic example of the BOS measurement collected by the CatWalk system—walking track test. The base of support (BOS) was used as an indirect measurement of proprioceptive recovery after sciatic nerve injury. (**D**) BOS comparison between the strains showed a greater compensation for the C57BL/6J mice when compared with the A/J, depicting a chronic insufficiency The SFI is expressed as a percentage of the preoperative value; the mean value of the nociceptive threshold is proportional to the control group; the BOS is measured by the CatWalk system distance unit *n* = 5 per group/day for each strain. Mean ± SEM; * *p* < 0.05.

**Table 1 cells-11-03710-t001:** Primary and secondary antibodies used.

	Antibody	Manufacturer	Host	Cat. Number	Concentration
Primary	MHC Class I H-2b/D/P/Q/w16	Bio-Rad	Rat	MCA2398	1:200
	Synaptophysin	Novus Biological	Rabbit	NBP2-25170	1:1000
	IBA1	Wako	Rabbit	019-19741	1:750
	GFAP	Abcam	Rabbit	AB7260	1:750
Secondary	Cy3 anti-rat	Jackson ImmunoResearch	Monkey	712-165-153	1:250
	Alexa Fluor^®^ 488 anti-rabbit	Jackson ImmunoResearch	Monkey	711-545-152	1:350
	Cy2 anti-rabbit	Jackson ImmunoResearch	Monkey	711-255-152	1:250

**Table 2 cells-11-03710-t002:** List of TaqMan assays (Life Technologies) used in RT–qPCR.

Gene	Cat. Number
trkb	Mm00435422_m1
cofilin	Mm03057591_g1
shp2	Mm00448434_m1
crmp2	Mm00515559_m1
gapdh	Mm99999915_g1

**Table 3 cells-11-03710-t003:** Relative gene expression (RT–qPCR) in microdissected motoneurons present in the dorsal lamina IX of Rexed. Unlesioned = control. 7 dal = 7 days survival time after unilateral sciatic nerve crush at mid-thigh level.

	C57BL/6J		A/J	
Gene	Unlesioned	7 dal	Unlesioned	7 dal
Crmp2	1.040	1.823	1.027	0.620
Cofilin	1.020	1.443	0.9733	0.863
Shp2	1.033	2.397	1.045	1.125
Trkb	1.045	1.073	1.083	0.520

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
