# Peer review of "The Time Course of MHC-I Expression in C57BL/6J and A/J Mice Correlates with the Degree of Retrograde Gliosis in the Spinal Cord following Sciatic Nerve Crush"

_cells, 2022, doi:10.3390/cells11233710_

Round 1

Reviewer 1 Report (Previous Reviewer 1)

Good work

Author Response

Thank you.

Reviewer 2 Report (Previous Reviewer 2)

The authors have been very responsive to the previous review and significantly modified the manuscript and improved on previous mismatches between results and the conclusions offered. At the same time, they now provide new data that significantly enhances overall interpretation. The new results demonstrate that MHC-I is preferentially expressed in astrocytes and that differs from previous published work suggesting expression in axotomized motoneurons and/or microglia. This finding is a new addition to current literature. The main conclusion is that the astrocytic reaction (including MHC-I) is accelerated and peaks higher in AJ mice compared to CB57BL/6J. This led to a new conclusion suggesting that increased expression of regenerative genes in C57BL/6J despite a lower regenerative capacity is mainly due to a mistiming between regeneration associated genes and trophism promoting regeneration perhaps due to the more blunted astrocyte reaction. To support this explanation the authors cited Oliveira, A.L.R.; Langone, F. Non-Neuronal Cells Are Not the Limiting Factor for the Low Axonal Regeneration 735 in C57BL/6J Mice. Brazilian J. Med. Biol. Res. 2000, 33, 1467–1475, doi:10.1590/S0100-879X2000001200011. However, in reading this article I could not find the evidence to support this proposition. Am I misreading this art of the discussion? Can the authors clarify more specifically how they reached this conclusion?

A couple of other interpretation issues require attention.

1) The authors should also be careful of non-conflating correlation with causality when comparing astrocyte reaction and synaptic plasticity around the motoneurons. This article and a previous one from the same group (Emirandetti, A.; Graciele Zanon, R.; Sabha, M.; de Oliveira, A.L.R. Astrocyte Reactivity Influences the Number of Presynaptic Terminals Apposed to Spinal Motoneurons after Axotomy. Brain Res. 2006, 1095, 35–42, doi:10.1016/j.brainres.2006.04.021) clearly show that astrocyte activation 1-week postinjury is higher in AJ mouse compared to C57BL/6J, but while in the previous article larger synaptic plasticity paralleled the larger astrocytic reaction, in the present results it does not.  This lack of replicability is concerning and remained unexplained. Moreover, in these studies correlative findings are usually interpreted as being cauasility linked. This is not on always warranted without further experimentation. The authors need to consider that one of the few articles that modified specifically the astrocytic response around axotomized motoneurons by deleting STAT3 signaling specifically in astrocytes showed that diminishing the astrocyte reaction resulted in a larger (not less) loss of synapses around the cell bodies of axotomized motoneurons (Tyzack, G. E., Sitnikov, S., Barson, D., Adams-Carr, K. L., Lau, N. K., Kwok, J. C., et al. (2014). Astrocyte response to motor neuron injury promotes structural synaptic plasticity via STAT3-regulated TSP-1 expression. Nat. Commun. 5:4294. doi: 10.1038/ncomms5294).

2) In the discussion the authors also suggest that “By assuming that C57BL/6J mice present decreased regenerative potential, our results are in line with the observations of MILETIC, MILETIC, (2008), who demonstrated BDNF upregulation together with TrkB. Such regulation indirectly causes loss of GABAergic inhibition, which is a protective mechanism following injury, avoiding excessive ??2+ intracellular levels.”  This interpretation is a big stretch of current data. First, the Miletic study is about KCC2 protein regulation on the membrane of dorsal horn neurons by TrkB mechanisms after chronic constriction injury specifically in males, while in the ventral horn KCC2 gene expression is downregulated in axotomized motoneurons (after crush or cut of the scitic nerve) by a TrkB independent and sex-independent mechanisms (Akhter ET, Griffith RW, English AW, Alvarez FJ. Removal of the Potassium Chloride Co-Transporter from the Somatodendritic Membrane of Axotomized Motoneurons Is Independent of BDNF/TrkB Signaling but Is Controlled by Neuromuscular Innervation. eNeuro. 2019 Oct 16;6(5):ENEURO.0172-19.2019. doi: 10.1523/ENEURO.0172-19.2019) and the total removal of KCC2 in axotomized motoneurons renders GABAergic mechanisms excitatory not inhibitory on axotomized motoneurons (Toyoda H, Ohno K, Yamada J, Ikeda M, Okabe A, Sato K, Hashimoto K, Fukuda A. Induction of NMDA and GABAA receptor-mediated Ca2+ oscillations with KCC2 mRNA downregulation in injured facial motoneurons. J Neurophysiol. 2003 Mar;89(3):1353-62. doi: 10.1152/jn.00721.2002). I suggest modifying or removing this interpretation that seems to be inaccurate with current literature on specifically axotomized motoneurons.

Minor points

1) Lines 20-22. I am not sure this sentence is a complete statement. Reword? :  “In addition, mRNA expression 20 of microdissected spinal motoneurons revealed an increase in cytoskeleton-associated molecules 21 (cofilin, shp2, and crmp2, *p<0.05) except for trkB in C57BL/6J mice.

2) lines 38-43 Citations missing? MHC I upregulation following axotomy has been shown in different instances, and comparison among mouse strains revealed an innate variation in MHC I expression, which was higher in C57BL/6J mice than in 129/SvJ siblings (citation?). In line with that, we have previously shown that A/J mice present the greatest postinjury expression of MHC I among the studied strains, which correlates with the increase in astroglia and microglial reactions in the spinal cord after peripheral axotomy (citation?).

3) Line 119 and many others needs fixing: “…the experimental groups are detailed in Error! Reference source not found..” A similar problem is found throughout the text. I am not sure what is the formatting issue here and what is missing.

In conclusion the authors have significant worked on this revision and this manuscript has significantly improved but a couple of interpretation issues linger and need to be corrected.

Author Response

Response to Reviewer #2

  • The main conclusion is that the astrocytic reaction (including MHC-I) is accelerated and peaks higher in AJ mice compared to CB57BL/6J. This led to a new conclusion suggesting that increased expression of regenerative genes in C57BL/6J despite a lower regenerative capacity is mainly due to a mistiming between regeneration associated genes and trophism promoting regeneration perhaps due to the more blunted astrocyte reaction. To support this explanation the authors cited Oliveira, A.L.R.; Langone, F. Non-Neuronal Cells Are Not the Limiting Factor for the Low Axonal Regeneration 735 in C57BL/6J Mice. Brazilian J. Med. Biol. Res. 2000, 33, 1467–1475, doi:10.1590/S0100-879X2000001200011. However, in reading this article I could not find the evidence to support this proposition. Am I misreading this art of the discussion? Can the authors clarify more specifically how they reached this conclusion?

Response: Thank you for the question. In the above-mentioned work, the transplantation of pre-degenerated sciatic nerve grafts from A/J, and C57BL/6J mice resulted in similar axonal regeneration, indicating that the nerve microenvironment of C57BL/6J strain supports the regenerative process adequately. Such a finding raised the hypothesis that a possible mismatch of the regenerative response and the trophic support could happen in C57BL/6J mice, which fits well with the current findings in the present manuscript.

  • The authors should also be careful of non-conflating correlation with causality when comparing astrocyte reaction and synaptic plasticity around the motoneurons. This article and a previous one from the same group (Emirandetti, A.; Graciele Zanon, R.; Sabha, M.; de Oliveira, A.L.R. Astrocyte Reactivity Influences the Number of Presynaptic Terminals Apposed to Spinal Motoneurons after Axotomy. Brain Res. 2006, 1095, 35–42, doi:10.1016/j.brainres.2006.04.021) clearly show that astrocyte activation 1-week postinjury is higher in AJ mouse compared to C57BL/6J, but while in the previous article larger synaptic plasticity paralleled the larger astrocytic reaction, in the present results it does not.  This lack of replicability is concerning and remained unexplained. Moreover, in these studies correlative findings are usually interpreted as being cauasility linked. This is not on always warranted without further experimentation. The authors need to consider that one of the few articles that modified specifically the astrocytic response around axotomized motoneurons by deleting STAT3 signaling specifically in astrocytes showed that diminishing the astrocyte reaction resulted in a larger (not less) loss of synapses around the cell bodies of axotomized motoneurons (Tyzack, G. E., Sitnikov, S., Barson, D., Adams-Carr, K. L., Lau, N. K., Kwok, J. C., et al. (2014). Astrocyte response to motor neuron injury promotes structural synaptic plasticity via STAT3-regulated TSP-1 expression. Commun. 5:4294. doi: 10.1038/ncomms5294).

Response: The concern regarding synaptophysin labeling has been a point of discussion in our group, and we thank the reviewer for bringing that up. There are some technical shortcomings that may contribute to such differences, including the brand and epitope used to raise the synaptophysin antibody, which had to be changed due to discontinuation by the manufacturer. Also, in the work by Emirandetti et al., the quantification of the labeling was concentrated on the immediate surface of the motoneurons. Herein, the overall picture has been used for the quantification, so that the neuropile labeling intensity was taken into account instead of the motoneuron coverage alone. We added the Tyzack et al. citation to the present version of the manuscript (ref. 43).

  • In the discussion the authors also suggest that “By assuming that C57BL/6J mice present decreased regenerative potential, our results are in line with the observations of MILETIC, MILETIC, (2008), who demonstrated BDNF upregulation together with TrkB. Such regulation indirectly causes loss of GABAergic inhibition, which is a protective mechanism following injury, avoiding excessive ??2+ intracellular levels.”  This interpretation is a big stretch of current data. First, the Miletic study is about KCC2 protein regulation on the membrane of dorsal horn neurons by TrkB mechanisms after chronic constriction injury specifically in males, while in the ventral horn KCC2 gene expression is downregulated in axotomized motoneurons (after crush or cut of the scitic nerve) by a TrkB independent and sex-independent mechanisms (Akhter ET, Griffith RW, English AW, Alvarez FJ. Removal of the Potassium Chloride Co-Transporter from the Somatodendritic Membrane of Axotomized Motoneurons Is Independent of BDNF/TrkB Signaling but Is Controlled by Neuromuscular Innervation. eNeuro. 2019 Oct 16;6(5):ENEURO.0172-19.2019. doi: 10.1523/ENEURO.0172-19.2019) and the total removal of KCC2 in axotomized motoneurons renders GABAergic mechanisms excitatory not inhibitory on axotomized motoneurons (Toyoda H, Ohno K, Yamada J, Ikeda M, Okabe A, Sato K, Hashimoto K, Fukuda A. Induction of NMDA and GABAA receptor-mediated Ca2+ oscillations with KCC2 mRNA downregulation in injured facial motoneurons. J Neurophysiol. 2003 Mar;89(3):1353-62. doi: 10.1152/jn.00721.2002). I suggest modifying or removing this interpretation that seems to be inaccurate with current literature on specifically axotomized motoneurons.

Response: We agree that the above-mentioned discussion is unnecessary so the phrase was removed.

  • Lines 20-22. I am not sure this sentence is a complete statement. Reword? :  “In addition, mRNA expression 20 of microdissected spinal motoneurons revealed an increase in cytoskeleton-associated molecules 21 (cofilin, shp2, and crmp2, *p<0.05) except for trkB in C57BL/6J mice.

Response: Thank you. We rephrased the sentence.

  • lines 38-43 Citations missing? MHC I upregulation following axotomy has been shown in different instances, and comparison among mouse strains revealed an innate variation in MHC I expression, which was higher in C57BL/6J mice than in 129/SvJ siblings (citation?). In line with that, we have previously shown that A/J mice present the greatest postinjury expression of MHC I among the studied strains, which correlates with the increase in astroglia and microglial reactions in the spinal cord after peripheral axotomy (citation?).

Response: We added the appropriate citations.

  • Line 119 and many others needs fixing: “…the experimental groups are detailed in Error! Reference source not found..” A similar problem is found throughout the text. I am not sure what is the formatting issue here and what is missing.

Response: Fixed.

This manuscript is a resubmission of an earlier submission. The following is a list of the peer review reports and author responses from that submission.

Round 1

Reviewer 1 Report

Authors should eliminate the last paragraphs of the introduction. results and conclusions are not part of the introduction.

The results section has many interpretations and comments that are specific to the discussion.

References used in the text should appear in chronological order.

The first two paragraphs of the discussion should be eliminated, they are part of the introduction. The discussion section should be rewritten and focused on the specific discussion of the results.

In sum, this manuscript is poorly written. 

Author Response

Response to Reviewer #1

Reviewer’s comments:

Authors should eliminate the last paragraphs of the introduction, results and conclusions are not part of the introduction. The results section has many interpretations and comments that are specific to the discussion. References used in the text should appear in chronological order. The first two paragraphs of the discussion should be eliminated, they are part of the introduction. The discussion section should be rewritten and focused on the specific discussion of the results. In sum, this manuscript is poorly written.

Response: Thank you for your comments. We eliminated the last paragraph of the introduction, and revised the entire text, including new data and the respective interpretation of the results. The manuscript has been revised by the American Journal Experts (please see the attached certificate), so that the quality of the text is improved. 

Reviewer 2 Report

The present article compares two strains of mice C56Bl/6J and A/J for their central neuroinflammatory reaction after peripheral nerve crush and the expression of synaptic plasticity and cytoskeletal genes related to possibly regeneration. The assumption is that all these are causally related and influence functional recovery. Thus, a final set of experiments compares the recovery of sensorimotor function estimated by the Sciatic Nerve Index, proprioceptive function estimated by the base support length in hindlimb steps and sensory recovery estimated by Von-Frey filament testing. The significance of this study is dual. First, correlative features of the glial reaction, synaptic plasticity, molecular changes in regenerative genes and functional recovery among the two mouse lines could be used to infer possible relationships. Second, this manuscript comparing mouse strains might suggest cellular mechanisms that differ according to genetic background to explain the different prognosis for recovery from similar nerve injuries within the human population. The manuscript contains data that seems obtained with high standards and the figures have high quality, however the interpretation and discussion of results have significant flaws, in my view. The authors may have stretched the data too much to fit old, entrenched assumptions. Therefore, this manuscript requires major revisions to adapt conclusions and logic to more realistic interpretations of the data presented. I will try to explain this in full in the following paragraphs.

The data shows that microglia MHC-I expression goes up in parallel in both strains, but with a significant higher expression at 7 days after injury then reduced by day 28. A parallel evolution was found for microglia Iba1 staining. This is kind of expected because MHC-I and Iba1 are both express in microglia which is well known to peak in proliferation at 7 days after crushing the sciatic nerve. Cell counts were not done, so it is not possible to interpret whether differences are due to microglia proliferation differences between both strains or differential regulation of MHC-I or Iba1 expression in microglia. This is just a minor point given the goals of the manuscript. One major point of concern is however the lack of Iba1-immunoreactivity in the unlesioned side (Figure 4).  Iba1 expression is found by all other authors in all microglia in a variety of activated states and also in surveillance mode. The authors should explain this discrepancy.

In contrast, GFAP expression peaked in both strains of mice at 5 days after injury and this should correspond to a change in expression in astrocytes, since this glial cell is known to not proliferate inside the spinal cord after a peripheral nerve crush. By difference to the obvious change in microglia the differences in GFAP between both strains occur at 5 days after injury and are subtle. This transient, early and small effect is of unclear biological relevance and the authors use of these data in discussion and interpretation as clear differences in glial responses seems an overreach.

No differences were found between both strains in synaptic plasticity around injured motoneurons estimated by synaptophysin densitometry. This suggests that differences encountered in the microglia reactions do not modify the intensity of synaptic plasticity around axotomized motoneuron. This conclusion is identical to that from Breg et al., and this previous paper needs to be referenced and discussed by the authors (Berg A, Zelano J, Thams S, Cullheim S. The extent of synaptic stripping of motoneurons after axotomy is not correlated to activation of surrounding glia or downregulation of postsynaptic adhesion molecules. PLoS One. 2013;8(3):e59647. doi: 10.1371/journal.pone.0059647. Epub 2013 Mar 19. PMID: 23527240; PMCID: PMC3602371.) Berg et al. used sciatic nerve transections compared different mouse strains and performed similar synaptophysin immunolabelings. Therefore, the explanation proposed in the discussion (page 19 paragraph 3), that the lack of correlation between glia reactions and synaptic plasticity found in the current manuscript is a peculiarity unique to nerve crush, is invalid and needs to be modified. The most current literature suggests that a clear relationship between glial and synaptic plasticity around axotomized motoneurons does not exist or is not as straightforward as previously assumed. The cited reports with “suggestions” along this line are based on experimental designs from which causal relationships could not be derived (correlations). When this relationship has been tested with adequate experimental designs it was always found that synaptic plasticity around axotomized motoneurons occurs quite independent of the microglia reaction, although a possible modulatory role has not been totally ruled out. Moreover, the astrocyte reaction is dependent on microglia activation and therefore preventing it reduces both, with no changes in synaptic plasticity.  The results presented here further support this view (no or little relationship between synaptic plasticity and glial reactions), however the authors remarks in the discussion keep underscoring the opposite (strong relationship). The papers demonstrating the lack of relationship between the microglia reaction and synaptic plasticity seem ignored.

In a following set of experiments, they report using RT-qPCR that the crmp2, cofilin and shp2 genes are more highly upregulated in C57BL/6j mice compared to A/J mice. If the data on crmp2, cofilin and shp2 is interpreted as molecular changes that enhance axonal regrowth and repair this would suggest that synaptic plasticity around motoneuron cell bodies has no influence on regenerative gene expression contradicting the first paragraph of the introduction. In addition, the higher expression of regenerative genes in C57BL/6J mice contradicts the presentation of C57Bl/6J mice as low regeneration potential. The introduction cites several papers that convincingly make the case that there are significant differences in axonal regeneration between C56BL/6J and A/J mice, but these papers also suggest that differences are likely due to a difference in the regrowth of sensory axons, while motor axons grow at similar rates. The authors should more carefully discuss these details to best interpret the present data.

Finally, the behavioral outcomes show similar recovery in SCI index and Von-Frey responses, with may be some small significant effects at isolated time points during the recovery time course. This contrast with the consistent literature on C57B/6J animals having slower regeneration.  The authors argue again that this is because differences between regeneration after crush or full nerve transection. It will be desirable to elaborate on possible mechanisms and further support this hypothesis.

Other issues that need to be addressed

1.                   Lindman, et al., Neuroimmunology 2002 cited in the first paragraph of the introduction does not show “a differential response of MHC-I in response to interferon beta administration” or shows that “the regenerative potential of lesioned neuron can be enhanced by removal of presynaptic input”. This paper is about the differential expression of MHC-I antigens and beta2-microglobulin mRNA in different strains of mice and the lack of effects of genetic deletion of interferon gamma or STAT4 or STAT6. Moreover, the only RAG gene tested, GAP43, did not differ among strains despite large differences in MHC-I upregulation. Please correct or provide a different reference.

2.                   Page 2. What does it mean that microglia has a “dubious behavior”? Please rephrase. Do you mean  dual …?

3.                   Please revise the whole paper for English language and remove words in Portuguese (Alemanha in page 4, disponivel em in reference list).

4.                   Alvarez and Cope papers cited in page 19 did not evaluate MHC I driven plasticity. The Alvarez group has examined microglia driven plasticity (not MHC-I driven plasticity) and the best reference is Rotterman et al 2019 which is the one testing causality between Ia afferent synaptic changes and the microglia reaction (Rotterman TM, Akhter ET, Lane AR, MacPherson KP, García VV, Tansey MG, Alvarez FJ. Spinal Motor Circuit Synaptic Plasticity after Peripheral Nerve Injury Depends on Microglia Activation and a CCR2 Mechanism. J Neurosci. 2019 May 1;39(18):3412-3433. doi: 10.1523/JNEUROSCI.2945-17.2019. Epub 2019 Mar 4. PMID: 30833511; PMCID: PMC6495126). Please correct.

5.                   A small presentation issue is that in figures 7 and 8 the panels containing the data for each individual strain in isolation seem unnecessary (A,B D, E in Figure 7; B, C in Figure 8). The data should be shown superimposed and if different sets of pairwise statistical comparison are done (in between strains and in single line vs control) the authors could use asterisks and hashtags for each type of comparison.

Author Response

Authors' responses are in the attached file.
